# Association ankle function and balance in community-dwelling older adults

**David Hernández-Guillén** [1]*, **Catalina Tolsada-Velasco**[2], **Sergio Roig-Casasús**[1,3], **Elena Costa-Moreno**[2], **Irene Borja-de-Fuentes**[2], **José-María Blasco**[1]

**1** Group of Physiotherapy in the Ageing Processes, Departament de Fisioteràpia, Universitat de València, Valencia, Spain, **2** Departament de Fisioteràpia, Universitat de València, Valencia, Spain, **3** Hospital Politècnic i Universitari La Fe, Valencia, Spain

* david.hernandez@uv.es

**Data Availability Statement:** All relevant data are within the paper and its Supporting Information files.

## Abstract

### Background and purpose

Ankle function declines with age. The objective of this study was to investigate the association between ankle function and balance in older adults, with a focus on range of motion (ROM) and strength.

### Methods

This was a cross-sectional study that included 88 healthy community-dwelling older adults. Ankle mobility was measured while bearing weight (lunge test) and not bearing weight. The plantar-flexor muscle strength was assessed using a hand-held dynamometer. Balance was measured in terms of dynamic balance and mobility (timed up and go test), monopodal and bipodal static balance with open and closed eyes (single-leg stand test and platform measures), and margins of stability (functional reach test). Linear correlation and multiple regression analyses were conducted with a 95% CI.

### Results and discussion

Most participants had limited ankle mobility (n = 75, 86%). Weight-bearing ankle dorsiflexion ROM was the strongest predictor of dynamic balance and included general mobility and stability ($R^2_{adj}$ = [0.34]; $\beta$ = [-0.50]). In contrast, plantar-flexor muscle strength was a significant predictor of static standing balance with open eyes ($R^2_{adj}$ = [0.16–0.2]; $\beta$ = [0.29–0.34]). Overall, weight-bearing ankle dorsiflexion ROM was a more representative measure of balance and functional performance; however, a non-weight-bearing mobility assessment provides complementary information. Therefore, both measures can be used in clinical practice.

### Conclusion

This study supports the concept that ankle mobility contributes to the performance of dynamic tasks, while the plantar-flexor muscle strength helps to develop a standing static balance. Identification of alterations in ankle function is warranted and may assist in the design of tailored interventions. These interventions can be used in isolation or to augment

**Funding:** The author(s) received no specific funding for this work.

**Competing interests:** The authors have declared that no competing interests exist.

conventional balance training in order to improve balance performance in community-dwelling older adults.

## Introduction

Feet are the base of support for the body, while the ankle plays a key role in primary movements [1, 2]. Therefore, its correct biomechanical functioning is necessary to perform activities of daily living [3]. However, loss of strength and limited ankle mobility are both common with aging. Flattening of the plantar arch, decrease in water and synovial fluid in the cartilage, and reduced quality of collagen cross-links, which increase stiffness, are among the age-related physiological and morphological factors that limit ankle function [4, 5].

Biomechanical constraints in the ankle joint can potentially contribute to decrease in balance performance in older adults. This, together with falls, are among the major concerns related to age [6], which impact over 25% of community-dwelling older adults [7]. Loss of ankle strength has been associated with altered postural sway, decreased walking speed, and reduced capacity to develop other functional movements [8–10]. In addition, ankle mobility has been suggested to be associated with postural control and may predict the risk of falls [4, 11]. It is well established that limited balance is a direct cause of falls and is a leading cause of morbidity and mortality. This results in social and economic costs for patients, families, and community services [7]. Therefore, approaches aimed at detecting factors that influence balance and falls are essential.

Previous research analyzed the association between several ankle and foot characteristics, including ankle function (mainly in terms of mobility and strength) with diverse balance components. However, these were conducted with varied methodologies, such as with populations different from older adults, such as young adults or athletes [12, 13]; with diverse conditions, such as stroke or ankle instability [14, 15]; and with inconsistent measurement methods, such as active, active-assistive, passive, weight, and non-weight-bearing ankle ROM measurements [5, 8, 16–19]. In addition, kinematic and kinetic measures, i.e., mobility and strength, may provide supplementary information. However, few studies included both of these in their methods to assess their influence on the balance of older adults [12, 17].

Given the above factors, the objective of this study was to investigate the association between the ankle (i.e. the tibiofibular-talar joint) function in the sagittal plane (plantar-flexor strength, and weight- and non-weight-bearing mobility) with postural control (monopedal and bipedal static balance, dynamic balance, mobility, and stability) in community-dwelling older adults. This is a population prone to limited balance performance and limited ankle function owing to the aging process [18, 20].

## Methods

This cross-sectional study complied with the Declaration of Helsinki and was approved by the ethics board of the University of Valencia (no. H1543937079194). The research was prospectively registered (NCT-03898999). Participants were recruited from four public associations and social facilities for seniors after obtaining written permission from the respective boards. The University of Valencia was responsible for the integrity and conduct of the study. Healthy volunteers willing to participate were included if they were independent community-dwelling older adults over 60 years with no history of lower limb injury in the 3 months prior to the study (e.g., sprain) or without a known pathology affecting balance (e.g., stroke, vestibular

affection, neuropathies, strong visual, or auditory limitations). All participants were informed verbally and in writing and signed a consent form to participate. Assessment of participants began in April 2019 and was concluded in May 2019. Compliance with the inclusion criteria, demographic information, ankle range of motion (ROM), and plantar-flexor strength were collected by the same experienced physiotherapist (>15 years of experience). Subsequently, two members of the team, blinded to the ankle function results, assessed the performance tests. Each test was assigned to one of the outcome assessors. Another member of the team was responsible for data analysis, for which participants were coded with numerical identifiers.

## Candidate regressors and data collection procedures

Each participant was assessed individually. A member of the team assessed the participant's ankle mobility and strength. Mobility was assessed in the sagittal plane, as this is where the main ROM occurs. Two major measurements were conducted, namely, an analytical assessment without weight-bearing, and a functional assessment, conducted while weight-bearing. Initially, the active non-weight-bearing ankle ROM was assessed using a telescopic goniometer. This was conducted with the participant laying in the supine position with a wedge under the knees to eliminate tension on the gastrocnemius muscles (ICC = 0.85–0.96) [21]. Subsequently, we used the lunge test [22] to measure the ankle dorsiflexion ROM by registering the maximum tilt of the tibia while standing and bearing weight on the limb without lifting the heel from the floor. A digital inclinometer was placed 15 cm below the anterior tuberosity of the tibia (ICC = 0.80–0.89) [23]. As for the kinetic function, the strength of the plantar-flexor musculature of the ankle joint was measured [3]. The participant was instructed to lie with the leg extended, and the measurement was taken using a hand-held dynamometer placed on the metatarsal head (ICC = 0.77–0.88) [24, 25].

As balance performance is usually also limited in older adults, this research looked for possible contribution of the ankle function parameters to altered balance. After assessing the ankle function, two independent researchers measured the balance performance of participants. Given the multifactorial nature of postural control, several tests usually related to various balance components were proposed. The tests were completed in the following order, allowing from 3 to 5 min of rest between tests: The open and closed-eye Romberg tests were used to assess static balance, with the participant standing upright on a T-Plate® pressure platform for 30 s [26]; the excursions from the center of pressure in terms of the swayed area ($mm^2$) and velocity ($mm^2/s$) were recorded. The single-leg stance test, a test used to assess monopodal stability, was used to estimate static balance and static stability; the participants were instructed to bear the weight on one limb and maintain their balance for a maximum of 60 s. The participants were allowed three attempts, and the maximum time was used for further analysis (ICC = 0.91–0.99) [27]. The functional reach test was used as a reasonable approximation of the margins of stability in the anterior-posterior direction; the participants were instructed to stand close to, but not touching, a wall where a meter was placed to measure their ability to reach distances. The arm was positioned with 90˚ of shoulder flexion and held with a closed fist. The starting position measurement was taken on the third metacarpal head. The participants were then instructed to reach as far as possible without taking a step, and the location of the third metacarpal head was subsequently recorded. (ICC = 0.92) [28]. The timed up and go test, a timed test of general mobility, was conducted to assess dynamic balance. The participants were instructed to get up from an armchair, walk 3 m, turn around, come back, and sit again. The mean of three attempts was used for further analysis (ICC = 0.97–0.99) [29]. Regular users of glasses or auditory devices performed the tests with their glasses or hearing aids, respectively, in order to diminish the impact of these common deficits on the balance results.

## Data analysis

Data were analyzed using SPSS software version 24.0 (IBM Corp., Armonk, NY, USA) licensed by the University of Valencia. A descriptive analysis of participants' characteristics was conducted. Data are presented as means and standard deviations, frequencies, and percentages. The normality of the distribution for the quantitative variables was verified using the Shapiro-Wilk test. The ankle ROM is presented as the average of both feet and represented in histograms. In addition, the descriptive synthesis classified the sample as an ankle mobility limitation when the dorsiflexion ROM in terms of the lunge test was <35˚, and the non-weight-bearing assessment was limited to ≤59˚ [30, 31].

The association between ankle function and postural control was analyzed using a two-level analysis. A multiple linear regression analysis using the stepwise method determined the factors that best predicted balance performance in older adults. The candidate regressors were selected considering significant correlations according to the *r* statistic, with 95% CIs. The analysis included balance performance measures (timed up and go, functional reach, single-leg stance, and posturography assessments) as the dependent variables and controlled for weight-bearing, ankle ROM, non-weight-bearing ankle ROM, and strength (independent variables). Demographic variables were also considered, as increasing age and weight could be expected to negatively affect balance, while a higher center of gravity is associated with lower stability (in physics terms). The models were compared using partial *F*-tests. The sample size for this study was determined a priori: A correlation of $r = 0.5$ and a power of 0.8, with $\alpha = 0,05$, estimated that 80 participants were required. Calculations were performed using the G*Power 3.1 software tool.

## Results

Eighty-eight participants were recruited, aged 75.2 years (SD 8.1), of which 64 (74%) were women. There was one dropout (one participant did not attend the scheduled assessment session). The analyzed sample was normally distributed in terms of ankle mobility, overall balance performance assessments, and plantar-flexor muscle strength. However, the sample was not normally distributed in terms of sex (more women) and in postural sway while standing in an upright position. The sample characteristics are listed in Table 1.

### Ankle range of motion in older adults

Most participants had limited ankle mobility in the sagittal plane in terms of weight-bearing dorsiflexion (n = 76, 87%) and non-weight-bearing ankle ROM (n = 75, 86%). The average ankle weight-bearing dorsiflexion ROM was below the threshold of 35˚ (27.3˚ [SD 7.1˚]) and of 59˚ while not bearing weight (49.8˚ [SD 9.9˚]). Although normally distributed, ankle mobility was highly variable among the participants and ranged from 6.9˚ to 43.2˚ when assessed in terms of the lunge test and from 26.5˚ and 76˚ when full ROM was assessed. Fig 1 illustrates the distribution.

### Factors predicting balance performance

Ankle mobility assessments presented a moderate correlation among the participants ($r = 0.42$; $p<0.001$), and these correlated with strength ($r>0.2$; $p<0.05$). The resultant models suggested that weight-bearing ankle mobility was a significant predictor of worse scores in the timed up and go test and functional reach test ($\beta = 0.5$; $p<0.001$). Non-weight-bearing mobility and plantar-flexor muscle strength also correlated with such balance outcomes but to a lesser extent ($r = 0.2–0.5$; $p<0.05$). However, there was no correlation between ankle mobility and static measures (Table 2).

**Table 1. Sample characteristics.**

| Sample characteristics | M (SD) | P-value* |
|---|---|---|
| n | 87 | - |
| Women (n, %) | 64 (74%) | - |
| Age (years) | 75.2 (8.1) | 0.10 |
| Height (cm) | 161.1 (8.5) | 0.20 |
| Weight (kg) | 69.2 (10.7) | 0.20 |
| **Ankle function** | | |
| Weight-bearing dorsiflexion ROM (degrees) | 27.3 (7.1) | 0.20 |
| Weight-bearing dorsiflexion ROM (n<35 degrees) | 76 (87%) | - |
| Non-weight bearing ROM (degrees) | 49.8 (9.9) | 0.20 |
| Non-weight bearing ROM (n<59 degrees) | 75 (86%) | - |
| Plantar-flexor strength (kg) | 15.8 (5.8) | 0.06 |
| **Balance performance** | | |
| One-leg single stance (s) | 20.6 (19.4) | 0.32 |
| Functional reach (cm) | 21.7 (5.9) | 0.20 |
| Timed up and go (s) | 10.3 (3.5) | 0.06 |
| Swayed area open eyes (mm$^2$) | 191.3 (469.3) | <0.001 |
| Velocity open eyes (ms$^{-1}$) | 4.5 (11.1) | <0.001 |
| Swayed area closed eyes (mm$^2$) | 157.2 (267.0) | <0.001 |
| Velocity closed eyes (ms$^{-1}$) | 4.9 (12.0) | <0.001 |

Note

* Results of normality test with Shapiro-Wilk ($p>0.05$ normal distribution)

As for plantar-flexor strength, a limitation in this measure was a significant predictor of the Romberg test score with open eyes ($R^2_{adj}$ = [0.16 to 0.20]). Finally, increasing age predicted worse scores in the timed up and go ($\beta$ = 0.24; $p<0.010$) and single-leg stance tests ($\beta$ = -0.48; $p<0.001$), while the participants' height influenced the functional reach results ($\beta$ = 0.20; p<0.037). The details are shown in Tables 2 and 3.

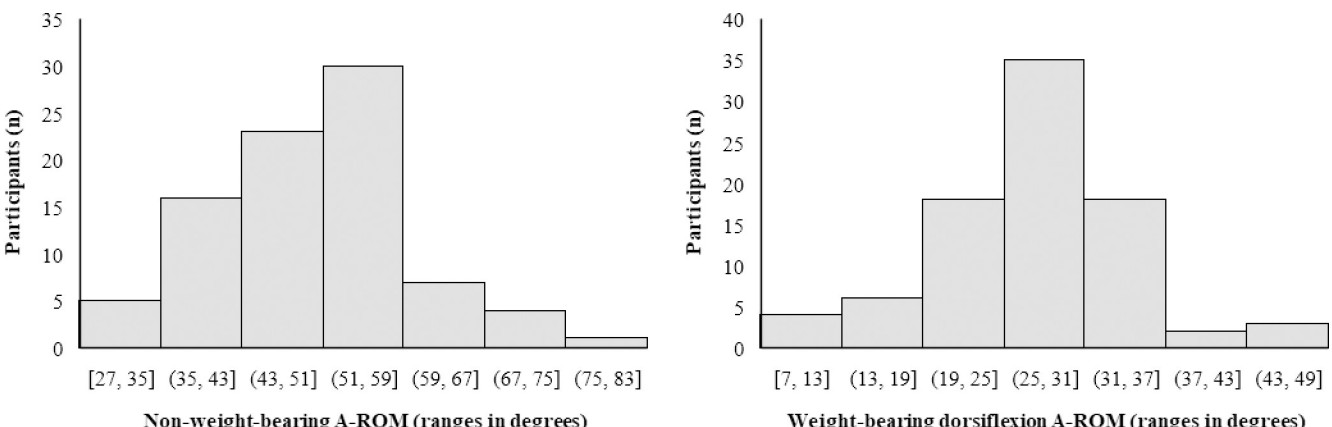

**Fig 1. Participants classified per ankle ROM (A-ROM) assessed under weight- and non-weight-bearing conditions.** Ankle mobility is considered to be limited in the sagittal plane when the weight-bearing dorsiflexion A-ROM in terms of the lunge test is less than 35˚, and the non-weight-bearing assessment is ≤59˚.

**Table 2. Association (Pearson's correlation *r*) between the candidate regressors (demographical characteristics and participants' ankle function) and balance performance.**

|  | Timed up and go | Single-leg stance | Functional reach | Swayed area OE | Velocity OE | Swayed area CE | Velocity CE |
|---|---|---|---|---|---|---|---|
| **Sample characteristics** | | | | | | | |
| Age | 0.34 $^{(*)}$ | -0.51 $^{(+)}$ | -0.23 $^{(*)}$ | 0.09 | 0.08 | 0.13 | 0.06 |
| Height | -0.11 | -0.00 | 0.32 $^{(**)}$ | 0.15 | 0.22 $^{(*)}$ | 0.18 | -0.01 |
| Weight | 0.03 | -0.13 | -0.02 | 0.11 | 0.16 | 0.11 | 0.02 |
| **Ankle function** | | | | | | | |
| Weight bearing DF-ROM | -0.55 $^{(+)}$ | 0.12 | 0.54 $^{(+)}$ | -0.07 | -0.04 | 0.04 | 0.05 |
| Non-weight bearing ROM | -0.30 $^{(**)}$ | 0.13 | 0.24 $^{(*)}$ | 0.16 | 0.13 | 0.15 | 0.14 |
| Plantar-flexor strength | -0.25 $^{(*)}$ | 0.05 | 0.30 $^{(**)}$ | 0.27 $^{(*)}$ | 0.28 $^{(**)}$ | 0.15 | 0.00 |

Abbreviations: DF: dorsiflexion; ROM: range of motion; OE/CE: Open/Closed eyes.

Notes

$^{(*)}$ *p*<0.05

$^{(**)}$ *p*<0.01

$^{(+)}$ *p*<0.001

## Discussion

Weight-bearing ankle dorsiflexion ROM had the strongest association value with general mobility and dynamic stability in older adults. This inferred that ankle mobility while bearing weight was an essential articular function to perform dynamic tasks. In contrast, the plantar-flexor muscle strength was a potential contributor to static standing balance with open eyes. Importantly, we found that a high proportion of participants (>80%) presented with limited ankle mobility. Given the above findings, further investigation is needed to determine whether treatment strategies aimed at restoring alterations in ankle function could improve balance, an ability that is commonly limited in older adults.

Weight- and non-weight-bearing ankle ROM measures were similarly associated with balance measures and, consistent with previous findings, had only moderate correlation in our study [32]. However, the former presented the highest correlation values and was the only predictive factor of balance according to statistical models. Probably because the ankle joint supports a large part of the weight of the human body and bears weight during dynamic functional tasks, such as walking, getting up from a chair, or climbing stairs. However, limited ankle ROM in the sagittal plane had little influence on maintaining static balance. This was most likely the case because static balance is not affected by minor ankle ROM limitations, and

**Table 3. Significant predictors of balance performance and parameters of the proposed prediction models.**

| Balance assessment | Measure | $R^2_{adj}$ | Predictor | B (SE) | β | P-value |
|---|---|---|---|---|---|---|
| General mobility / Dynamic balance | Timed up and go | 0.338 | Weight bearing DF-ROM | -0.25 (0.05) | -0.50 | <0.001 |
|  |  |  | Age | 0.10 (0.04) | 0.24 | 0.010 |
| Limits of stability | Functional reach | 0.316 | Weight bearing DF-ROM | 0.41 (0.07) | 0.50 | <0.001 |
|  |  |  | Height | 0.14 (0.06) | 0.20 | 0.037 |
| Monopodal stability | Single-leg stance | 0.301 | Age | -1.16 (0.22) | -0.48 | <0.001 |
| Bipodal stability / Static standing balance | Swayed area OE | 0.200 | Plantar-flexor strength | 28.24 (8.10) | 0.34 | 0.001 |
|  | Velocity OE | 0.160 | Plantar-flexor strength | 0.60 (0.19) | 0.29 | 0.007 |

Abbreviations: DF: dorsiflexion; ROM: range of motion; OE/CE: Open/Closed

is more dependent on multiple other factors, such as proprioception, vestibular function, sensory integration, or even vision. If these senses are healthy, the role of ankle ROM in static balance is limited and older adults are able to compensate for the vast majority of musculoskeletal limitations, thus avoiding losing their balance [3].

The role played by ankle mobility in postural control has been broadly discussed in the literature [3]. However, few studies have thoroughly appraised how this factor influences the various components of postural control or classified balance assessments in terms of measures of stability, dynamic, and static balance. Mecagni *et al.* [18] found positive associations among ankle mobility, functional mobility, and stability. Menz *et al.* [8] suggested that foot problems (including ankle ROM) may decrease stability and functional ability (consistent with our findings) and influence standing static balance (this was non-consistent). Finally, Spink *et al.* [17] found that ankle inversion-eversion ROM was an important determinant of balance in older adults (our study evaluated dorsiflexion to plantarflexion alone). The present study reinforced ankle mobility as a potential contributor to dynamic balance performance, but not static balance.

As for the plantar-flexor muscle strength, the results suggested that, to some extent, this was a potential contributor to static standing balance. This reinforces previous studies supporting the fact that an increase in muscle strength may induce benefits on balance [33]. However, this relationship was only present when the participant's eyes were open. The elimination of an important sensory input, such as vision, results in greater reliance on proprioception, as well as on the integrative, vestibular, and underlying systems of balance. Whether the plantar-flexor strength becomes less relevant in the response strategies of these systems is speculative. Some studies have indicated that not only the maximal strength, but also aspects such as the capability of rapid strength production of plantar flexion, is important for balance ability in older adults [34].

From a clinical perspective, the literature has established key aspects for balance training in older adults [6], with exercises that train proprioception, challenge balance, increase lower limb and core musculature strength, reduce the support base, or even make it unstable. In addition, we support that ankle function had a direct influence on balance, although most older adults presented with reduced ankle mobility. Therefore, identifying this alteration may help to adapt and tailor conventional balance interventions, either by augmenting treatments or including exercises oriented at increasing ankle ROM to enhance clinical benefits. Among the possibilities, stretching and manual therapy are effective, low-cost, and low-implementation time approaches to be used in the older adult population [35, 36].

Overall, the importance of this study is to support the fact that ankle function may predict balance performance in older adults. Increasing age involved worse balance, but this was a significant predictor in only a few of the models. Height determined the limits of stability, as expected. The multi-component nature of balance has been confirmed once again. This study suggests the advisability of assessing ankle dorsiflexion mobility while bearing weight due to its stronger association with most aspects of postural control. However, weight- and non-weight-bearing assessments provide complementary information; therefore, both should be implemented in clinical practice whenever possible.

Some limitations of this study should be acknowledged and may guide future research. Ankle ROM was evaluated in the sagittal plane, while the main movement ranged from dorsiflexion to plantarflexion. This joint may move in other planes and it remains to be determined how other important movements (inversion and eversion) influence postural control in older adults; therefore, the findings of this study did not cover the prediction of dynamic balance ability in the medio-lateral direction, which may also be a common direction of falling. Although it has been shown to be a reliable measure, the use of the functional reach test to

assess stability remains controversial [28]. Considering that a large proportion of the included community-dwelling older adults presented with some degree of ankle mobility limitation, a future study analyzing individuals across the lifespan is of interest.

## Conclusion

This study showed that ankle function is associated with balance performance in community-dwelling older adults. Weight-bearing ankle dorsiflexion ROM in the sagittal plane is a potential predictor of dynamic balance performance. Consequently, individuals with higher mobility perform better in dynamic tasks. However, the plantar-flexor muscle strength was a significant predictor of static standing balance.

## Supporting information

**S1 Checklist. STROBE statement—checklist of items that should be included in reports of observational studies.**
(DOC)

**S1 Data set.**
(XLSX)

## Author Contributions

**Conceptualization:** David Hernández-Guillén, Sergio Roig-Casasús, José-María Blasco.

**Data curation:** David Hernández-Guillén, Catalina Tolsada-Velasco, Elena Costa-Moreno, José-María Blasco.

**Formal analysis:** David Hernández-Guillén, Sergio Roig-Casasús, Irene Borja-de-Fuentes, José-María Blasco.

**Investigation:** David Hernández-Guillén, Catalina Tolsada-Velasco, Sergio Roig-Casasús, Irene Borja-de-Fuentes, José-María Blasco.

**Methodology:** David Hernández-Guillén, Catalina Tolsada-Velasco, Elena Costa-Moreno, José-María Blasco.

**Project administration:** David Hernández-Guillén, Sergio Roig-Casasús, José-María Blasco.

**Resources:** David Hernández-Guillén, Elena Costa-Moreno, Irene Borja-de-Fuentes, José-María Blasco.

**Software:** David Hernández-Guillén, José-María Blasco.

**Supervision:** David Hernández-Guillén, Elena Costa-Moreno, José-María Blasco.

**Validation:** David Hernández-Guillén, Catalina Tolsada-Velasco, Elena Costa-Moreno, José-María Blasco.

**Visualization:** David Hernández-Guillén, Sergio Roig-Casasús, Elena Costa-Moreno, José-María Blasco.

**Writing – original draft:** David Hernández-Guillén, José-María Blasco.

**Writing – review & editing:** David Hernández-Guillén, Sergio Roig-Casasús, Elena Costa-Moreno, Irene Borja-de-Fuentes, José-María Blasco.

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
