## [Decision Letter · Decision Letter 0]

25 Nov 2020

PONE-D-20-30676

Association between Ankle Function and Balance in Community-Dwelling Older Adults

PLOS ONE

Dear Dr. Hernández-Guillén,

Thank you for submitting your manuscript to PLOS ONE. After careful consideration, we feel that it has merit but does not fully meet PLOS ONE’s publication criteria as it currently stands. Therefore, we invite you to submit a revised version of the manuscript that addresses the points raised during the review process.

Both reviewers assessed the manuscript quite positively, however, both also pointed out a number of points where information should be added or clarified. Also, the English writing is not up to standard. Please seek help to improve the language and to more clearly express your statements. 

We look forward to receiving your revised manuscript.

Kind regards,

Peter Andreas Federolf

Academic Editor

PLOS ONE

Journal Requirements:

Reviewers' comments:

Reviewer's Responses to Questions

**Comments to the Author**

1. Is the manuscript technically sound, and do the data support the conclusions?

Reviewer #1: Partly

Reviewer #2: Yes

2. Has the statistical analysis been performed appropriately and rigorously? 

Reviewer #1: Yes

Reviewer #2: Yes

3. Have the authors made all data underlying the findings in their manuscript fully available?

Reviewer #1: Yes

Reviewer #2: Yes

4. Is the manuscript presented in an intelligible fashion and written in standard English?

Reviewer #1: Yes

Reviewer #2: No

5. Review Comments to the Author

Reviewer #1: Thank you for sending me this manuscript. After thoroughly review it, some concerns would be clarified as follows:

Abstract

1. please add some magnitude of effects of the results the strongest predictor of dynamic balance.

Introduction

1. Please add other reasons for studying from a few of literatures evaluating relationship between kinematic and kinetic outcomes, and balance ability in the elderly. Does it have some conflicted evidence in this issue or what new information is added of this manuscript.

2. Change “goal” to “objective or aim” is more suitable.

Methods

1. Please provide information of these: study design, setting area of the study, recruitment method and period.

2. To include the eligible participants, do the participants should be active older adults?

3. Do other conditions such as neurological deficits DM with peripheral neuropathy, visual or auditory impairment or regular performing balance training strongly affect the outcomes of this study?

4. Please provide results of inter-rater reliability of two assessors.

5. Why do not the authors measure ankle inversion and eversion because such movements may influence balance ability in mediolateral direction of the elderly?

6. It seems that almost all parameters are lacking for details of testing session and representative values for statistical analysis, please add them.

7. Why do the authors only measure plantarflexors strength rather than measuring all ankle muscles?

8. It is required to add more information of balance testing that readers can do them accurately.

Discussion

1. Please give more reason why the results of the current study find that ankle mobility is a potential contributor to dynamic balance but not static balance.

2. I am not sure that sentences in Lines 225-227 are contrast together. Further, please provide more discussion of why plantarflexors strength show a high correlation with static balance.

3. According to lacking of other ankle movements such as inversion and eversion, the authors should note that the findings of this study would not cover the prediction of dynamic balance ability in mediolateral direction (may be a common direction of falling in the elderly).

Reviewer #2: This study examines the ankle factors associated with static and dynamic balance in a sample of 87 older adults, average age 75. The main finding is that plantar flexion strength is associated with static balance and ankle ROM (weight bearing) is associated with dynamic balance. The study has a well-defined aim, appears to be well-planned and executed, analysis is done elegantly. However, the manuscript requires extensive editing of grammar and sentence structure to improve readability. The manuscript presents several relevant points in the discussion that gets lost due to poor language.

Introduction: It would be good to highlight the incidence of falls in older adults, its implications (associated mortality, morbidity, financial implications) so as to highlight the public health relevance of the topic. Additionally, more substantial information on the role of balance in falls among older adults is warranted. (What proportion may be falling because of balance issues?)

Methods:

Page 4, Line 56: Please provide more detail on the “associations of older adults” from which participants were recruited so that readers have some information of the selected population and how they may be different from the general population of older adults in the community.

Data analysis:

Please specify clearly, the covariates used in each of the regression models. Were age, height and weight used in all the models? When weight bearing ROM was tested as a predictor, was the model adjusted for plantar flexion strength?

Page 5 Line 102: What was the two-level analysis that was done?

Discussion: There are multiple grammatical errors and poor sentence structure which makes the discussion difficult to read and understand. Please have a language editor edit the entire manuscript to improve the language.

It would be good to be more specific the clinical implications of the findings. What is the current clinical practice for therapy of older adults with poor balance? What are the muscle groups that are being focused on currently? What additional assessment and specific therapy would the authors recommend, given the findings?

6. PLOS authors have the option to publish the peer review history of their article (what does this mean?). If published, this will include your full peer review and any attached files.

Reviewer #1: No

Reviewer #2: No

---

## [Author Response · Author response to Decision Letter 0]

20 Jan 2021

Manuscript number: PONE-D-20-30676

Dear editor and reviewers

First, we would like to thank the time spent in carrying out a thorough review of our manuscript. We have studied the comments and suggestions. They are reasonable, and we do believe that they will increase the quality of the manuscript if we address them correctly,

In the following lines we respond point by point the considerations and suggestions made by reviewers and the amendments made by the authors to address these in the manuscript. 

The new submission includes a document with the manuscript in which all the changes introduced by the authors have been indicated with Word track changes, as well as another document with the final and clean version. This last version includes the corrections after English editing.

Reviewer #1: 

Thank you for sending me this manuscript. After thoroughly review it, some concerns would be clarified as follows:

Abstract

1. please add some magnitude of effects of the results the strongest predictor of dynamic balance.

a. Ok, we have included the information about the strongest predictor as suggested.

Introduction

1. Please add other reasons for studying from a few of literatures evaluating relationship between kinematic and kinetic outcomes, and balance ability in the elderly. Does it have some conflicted evidence in this issue or what new information is added of this manuscript.

a. Biomechanical analyses can rely either on kinetic, kinematic or both parameters. With this sentence we just wanted to emphasize that few studies considered both types in the same analysis. We have amended this paragraph and added some information for clarity

2. Change “goal” to “objective or aim” is more suitable.

a. Ok, done

Methods

1. Please provide information of these: study design, setting area of the study, recruitment method and period.

a. As suggested, we have included the following information within the text (please, kindly check the first paragraph of methods section):

i. Study design (cross-sectional) 

ii. Participants were recruited from four public associations and social facilities for seniors after obtaining written permission from the respective boards

iii. Assessment of participants began in April 2019 and was concluded in May 2019. 

2. To include the eligible participants, do the participants should be active older adults?

a. Indeed. Although we did not set any specific criteria regarding level of physical activity, we established that eligible participants had to be healthy community-dwelling older adults (and therefore, non-dependent and capable of doing their daily activities) with any known pathology that could affect balance (i.e. vestibular, neurological, etc). We have clarified this in the new manuscript.

3. Do other conditions such as neurological deficits DM with peripheral neuropathy, visual or auditory impairment or regular performing balance training strongly affect the outcomes of this study?

a. We need to consider that most older adults use to have some visual or auditory limitation. So, originally, these were not considered as an inclusion/exclusion criterion. Even though, none of the included participants presented with a strong visual or auditory deficit that precluded test performance. In addition, the participants were allowed to perform the test using their glasses or auditory devices, if they needed so (This information has been included in the end of data collection section and assessment section). However, neuropathies, vestibular disorders, neurological conditions were considered causes for exclusion, as has been detailed in the text.

b. On the other hand, we did not collect information about previous level of physical activity, since we did not consider to control this parameter. The reason is that it can be assumed that active people will likely present higher balance abilities and probably better joint mobility and vice versa. So, from our point of view, to control this factor or use it as an inclusion criterion could have been counterproductive, since can provide important information in the association analyses. Thank you for your comment.

4. Please provide results of inter-rater reliability of two assessors.

a. This is an important point. Indeed, three independent researchers assessed participants. The first one assessed ankle function and collected basic demographic/clinical data. To avoid bias, two other assessors conducted performance tests. However, each one was assigned with different tests, so ICC calculation was not needed, because the assessors were in charge to assess different outcome measures. This has been also explained in the text for clarity. Thank you for noticing. 

5. Why do not the authors measure ankle inversion and eversion because such movements may influence balance ability in mediolateral direction of the elderly?

a. Good point. If we may, our research targeted the tibiofibular-talar joint to assess how its mobility associated with balance. This joint mainly develops its motion in the sagittal plane. In addition, as the reviewer noticed, a limitation in the inversion-eversion movements had already been shown to negatively affect the balance of the elderly (Spink 2011). However, we need to consider that inversion-eversion movements mainly occur in the subtalar joint, also helped by the midfoot and forefoot, the contribution of the tibiofibular joint being small in this regard. Therefore, we originally decided not to assess this movement, because it is not an exclusive movement of the joint we targeted in this study. However, to address the reviewer suggestion, we have decided to recognize this as a possible limitation (kindly check limitations section). We hope this point has been clarified.

6. It seems that almost all parameters are lacking for details of testing session and representative values for statistical analysis, please add them.

a. As suggested, the measurement methods have been elaborated. This include: available psychometric information (i.e. ICC), the order in which the tests were performed, and a more detailed description on the tests. We believe that this section is now much more clear, thank you for your suggestion.

7. Why do the authors only measure plantarflexors strength rather than measuring all ankle muscles?

a. We decided to assess the strength of the plantar flexor musculature because previous studies suggested that this factor could negatively influence balance (Menz 2005, Spink 2011). Indeed, to assess the dorsiflexors strength would likely have provided important information, but we did not include this measurement. The assessment of the musculature that helps to carry out the inversion and eversion was discarded for the same reasons that have been argued in point 5. We hope that we have contributed to clarifying the doubts.

8. It is required to add more information of balance testing that readers can do them accurately.

a. This information has been elaborated as suggested, which has increased the quality of the presentation, so thank you for your comment. 

Discussion

1. Please give more reason why the results of the current study find that ankle mobility is a potential contributor to dynamic balance but not static balance.

a. We speculate whether the main reason would be that there is no need to perform a great ankle mobility to keep a static standing posture. Also, to be in upright standing is highly dependent of the correct functioning of the proprioceptive systems, the underlying mechanisms of balance, vestibular system, etc. These arguments have been further elaborated in the discussion section, in order to address the reviewer suggestion.

2. I am not sure that sentences in Lines 225-227 are contrast together. Further, please provide more discussion of why plantarflexors strength show a high correlation with static balance.

a. This paragraph (fourth in the discussion) has been elaborated and further contextualized with other literature.

3. According to lacking of other ankle movements such as inversion and eversion, the authors should note that the findings of this study would not cover the prediction of dynamic balance ability in mediolateral direction (may be a common direction of falling in the elderly).

a. We have pointed this as an additional limitation of our study. Thank you for your review work.

Reviewer #2: 

This study examines the ankle factors associated with static and dynamic balance in a sample of 87 older adults, average age 75. The main finding is that plantar flexion strength is associated with static balance and ankle ROM (weight bearing) is associated with dynamic balance. The study has a well-defined aim, appears to be well-planned and executed, analysis is done elegantly. However, the manuscript requires extensive editing of grammar and sentence structure to improve readability. The manuscript presents several relevant points in the discussion that gets lost due to poor language.

• Thank you for your suggestions. The manuscript has been sent to be copyedited by a certified English editing company. As non-native speakers, we can only hope that, after this action, the highest language standards are present in the manuscript. In addition, we respond point by point to the reviewer comments and suggestions:

Introduction: It would be good to highlight the incidence of falls in older adults, its implications (associated mortality, morbidity, financial implications) so as to highlight the public health relevance of the topic. Additionally, more substantial information on the role of balance in falls among older adults is warranted. (What proportion may be falling because of balance issues?)

• We have included new information on how balance decrease with age, and how limited balance affects falls in old people in the second paragraph, as suggested.

Methods:

Page 4, Line 56: Please provide more detail on the “associations of older adults” from which participants were recruited so that readers have some information of the selected population and how they may be different from the general population of older adults in the community.

• These were associations /social facilities for seniors, all of public funding, which are commonly distributed in every neighborhood of cities or villages (at least in Spain), all with very similar characteristics, for being supported with public funds. We have proceeded to explain this in the methods (first paragraph); hopefully this is clear now.

Data analysis:

Please specify clearly, the covariates used in each of the regression models. Were age, height and weight used in all the models? When weight bearing ROM was tested as a predictor, was the model adjusted for plantar flexion strength?

• As elaborated, we intended to test all the proposed candidate regressors. However, only those with significant association (based on r statistic, see table 2) were included in the models before start the iterations, while only the ones shown in the results were the candidate survivors that contribute to such models

Page 5 Line 102: What was the two-level analysis that was done?

• First, we studied the level of association based on r statistic. Second, those candidates that were significantly correlated (r>0.5; p<0.05, see table 2) were included in to test their prediction capability within the regression analysis and resultant model. This is explained in data analysis section; we hope it is more clear now.

Discussion: There are multiple grammatical errors and poor sentence structure which makes the discussion difficult to read and understand. Please have a language editor edit the entire manuscript to improve the language. 

• The reviewer is right, that is why we decided to send the paper to be proofread, thank you for noticing

It would be good to be more specific the clinical implications of the findings. What is the current clinical practice for therapy of older adults with poor balance? What are the muscle groups that are being focused on currently? What additional assessment and specific therapy would the authors recommend, given the findings?

• Thank you for your comment and your review work. All the suggested information has been included in a specific paragraph in the discussion section, specifically the fifth paragraph which starts with: From a clinical perspective…

---

## [Decision Letter · Decision Letter 1]

9 Feb 2021

PONE-D-20-30676R1

Association Ankle Function and Balance in Community-dwelling Older Adults

PLOS ONE

Dear Dr. Hernández-Guillén,

Thank you for submitting your manuscript to PLOS ONE. After careful consideration, we feel that it has merit but does not fully meet PLOS ONE’s publication criteria as it currently stands. Therefore, we invite you to submit a revised version of the manuscript that addresses the points raised during the review process.

Specifically, both reviewers were satisfied with the revisions. However, reviewer 2 still has several question/suggestions for language editing. Please consider these points.

We look forward to receiving your revised manuscript.

Kind regards,

Peter Andreas Federolf

Academic Editor

PLOS ONE

Reviewers' comments:

Reviewer's Responses to Questions

**Comments to the Author**

1. If the authors have adequately addressed your comments raised in a previous round of review and you feel that this manuscript is now acceptable for publication, you may indicate that here to bypass the “Comments to the Author” section, enter your conflict of interest statement in the “Confidential to Editor” section, and submit your "Accept" recommendation.

Reviewer #1: All comments have been addressed

Reviewer #2: All comments have been addressed

2. Is the manuscript technically sound, and do the data support the conclusions?

Reviewer #1: Yes

Reviewer #2: Yes

3. Has the statistical analysis been performed appropriately and rigorously? 

Reviewer #1: Yes

Reviewer #2: Yes

4. Have the authors made all data underlying the findings in their manuscript fully available?

Reviewer #1: Yes

Reviewer #2: Yes

5. Is the manuscript presented in an intelligible fashion and written in standard English?

Reviewer #1: Yes

Reviewer #2: Yes

6. Review Comments to the Author

Reviewer #1: (No Response)

Reviewer #2: The authors have revised the manuscript considerably, the resulting revised version has better clarity and is easier to read. The discussion section still has some language issues, and can be made more comprehensible with some more language editing.

Page 10 line 221: “only predictive balance assessment according to resultant models” - Do the authors mean that this was the “only predictive factor of balance according to statistical models”?

Page 10 line 222: “ proposed ankle mobility measures only had a moderate association” - It is difficult to understand that the authors are talking about the low correlation between weight bearing and non weight bearing ROM because of the terms “proposed” and “association”. Remove “in fact”, and simply state that the “weight bearing and non-weight bearing ROM had only moderate correlation in our study, consistent with previous findings”.

Page 10 line 223: “A possible explanation is that” - What does this explanation pertain to? The previous sentence talks about poor correlation between weight bearing and non weight bearing ROM, so the reader is expecting that the explanation is about that. As it is not, the authors have to state what the explanation is about at this point.

Page 10 line 223-234: It is not easy to understand the “explanation”. Do the authors mean that weight bearing on the ankle is an important part of dynamic functional tasks, hence it is logical that weight bearing ROM becomes a determinant of dynamic balance? On the other hand, static balance depends on multiple other factors and not just ankle function, hence the role of ankle ROM in static balance is limited? Please re-write to allow the reader to follow the train of thought.

Page 10 line 225: The authors mention static balance here along with dynamic, but in line 229, go on to say that ankle ROM is not important for static balance. This is contradictory and confusing.

Page 10 line 231: “does not require a great ROM” - Do the authors mean “ standing is not affected by minor ROM limitations at the ankle joint”?

Page 11 line 247: “Increase in musculature” - Do the authors mean increase in muscle strength, power or muscle mass?

Page 11 line 248: “finding was only evident” – Do the authors mean that the relationship was present only when the eyes were open?

Page 11 line 249: “implies greater confidence not only on balance” - Do the authors mean the elimination of vision results in greater reliance on proprioception for balance?

Please use simple and direct language for easy reading.

7. PLOS authors have the option to publish the peer review history of their article (what does this mean?). If published, this will include your full peer review and any attached files.

Reviewer #1: No

Reviewer #2: No

---

## [Author Response · Author response to Decision Letter 1]

15 Feb 2021

Manuscript number: PONE-D-20-30676-R2

Dear editor and reviewers

First, we would like to thank the time spent in carrying out this second review of our manuscript. We have studied the comments and suggestions. They are reasonable, and we do believe that they will increase the quality of the manuscript if we address them correctly,

In the following lines we respond point by point the considerations and suggestions made by reviewers and the amendments made by the authors to address these in the manuscript. 

The new submission includes a document with the manuscript in which all the changes introduced by the authors have been indicated with Word track changes, as well as another document with the final and clean version. This last version includes the corrections after English editing.

Reviewer #1: (No Response)

Reviewer #2: 

The authors have revised the manuscript considerably, the resulting revised version has better clarity and is easier to read. The discussion section still has some language issues, and can be made more comprehensible with some more language editing.

• Page 10 line 221: “only predictive balance assessment according to resultant models” - Do the authors mean that this was the “only predictive factor of balance according to statistical models”? � That was the meaning indeed. The sentence has been amended.

• Page 10 line 222: “proposed ankle mobility measures only had a moderate association” - It is difficult to understand that the authors are talking about the low correlation between weight bearing and non weight bearing ROM because of the terms “proposed” and “association”. Remove “in fact”, and simply state that the “weight bearing and non-weight bearing ROM had only moderate correlation in our study, consistent with previous findings” � Ok, amended as suggested

• Page 10 line 223: “A possible explanation is that” - What does this explanation pertain to? The previous sentence talks about poor correlation between weight bearing and non weight bearing ROM, so the reader is expecting that the explanation is about that. As it is not, the authors have to state what the explanation is about at this point � The reviewer is right. We have changed the order of the sentences in this paragraph so that the ‘possible explanation’ is directly related to the previous sentence. The text is more clear now, thank you for noticing.

• Page 10 line 223-234: It is not easy to understand the “explanation”. Do the authors mean that weight bearing on the ankle is an important part of dynamic functional tasks, hence it is logical that weight bearing ROM becomes a determinant of dynamic balance? On the other hand, static balance depends on multiple other factors and not just ankle function, hence the role of ankle ROM in static balance is limited? Please re-write to allow the reader to follow the train of thought. � As all previous comments were related to the same paragraph, this has been rewritten for clarity. We hope it reads better now. 

• Page 10 line 225: The authors mention static balance here along with dynamic, but in line 229, go on to say that ankle ROM is not important for static balance. This is contradictory and confusing � Indeed, this was a mistake that has been amended. As mentioned, the paragraph has been rewritten without losing the previous meaning.

• Page 10 line 231: “does not require a great ROM” - Do the authors mean “standing is not affected by minor ROM limitations at the ankle joint”? � Changed

• Page 11 line 247: “Increase in musculature” - Do the authors mean increase in muscle strength, power or muscle mass? � muscle strength. This has been corrected

• Page 11 line 248: “finding was only evident” – Do the authors mean that the relationship was present only when the eyes were open? � This was the case: changed

• Page 11 line 249: “implies greater confidence not only on balance” - Do the authors mean the elimination of vision results in greater reliance on proprioception for balance? � Corrected

• Please use simple and direct language for easy reading � The whole discussion section has been revised to address this comment, thank you for your review work.

---

## [Editor Report · Decision Letter 2]

16 Feb 2021

Association Ankle Function and Balance in Community-dwelling Older Adults

PONE-D-20-30676R2

Dear Dr. Hernández-Guillén,

We’re pleased to inform you that your manuscript has been judged scientifically suitable for publication and will be formally accepted for publication once it meets all outstanding technical requirements.

Kind regards,

Peter Andreas Federolf

Academic Editor

PLOS ONE
---

## [Editor Report · Acceptance letter]

23 Feb 2021

PONE-D-20-30676R2 

Association Ankle Function and Balance in Community-dwelling Older Adults 

Dear Dr. Hernández-Guillén:

I'm pleased to inform you that your manuscript has been deemed suitable for publication in PLOS ONE. Congratulations! Your manuscript is now with our production department. 

Kind regards, 

on behalf of

Dr. Peter Andreas Federolf 

Academic Editor

PLOS ONE